# Functionally antagonistic polyelectrolyte for electro-ionic soft actuator

Van Hiep Nguyen[1], Saewoong Oh[1], Manmatha Mahato [1], Rassoul Tabassian [1,2], Hyunjoon Yoo[1], Seong-Gyu Lee[3], Mousumi Garai[1], Kwang Jin Kim [4] & Il-Kwon Oh [1] ✉

Electro-active ionic soft actuators have been intensively investigated as an artificial muscle for soft robotics due to their large bending deformations at low voltages, small electric power consumption, superior energy density, high safety and biomimetic self-sensing actuation. However, their slow responses, poor durability and low bandwidth, mainly resulting from improper distribution of ionic conducting phase in polyelectrolyte membranes, hinder practical applications to real fields. We report a procedure to synthesize efficient polyelectrolyte membranes that have continuous conducting network suitable for electro-ionic artificial muscles. This functionally antagonistic solvent procedure makes amphiphilic Nafion molecules to assemble into micelles with ionic surfaces enclosing non-conducting cores. Especially, the ionic surfaces of these micelles combine together during casting process and form a continuous ionic conducting phase needed for high ionic conductivity, which boosts the performance of electro-ionic soft actuators by 10-time faster response and 36-time higher bending displacement. Furthermore, the developed muscle shows exceptional durability over 40 days under continuous actuation and broad bandwidth below 10 Hz, and is successfully applied to demonstrate an inchworm-mimetic soft robot and a kinetic tensegrity system.

Artificial muscle, other name of soft actuators, is a core component in soft robots[1–5], which can properly adapt to the unexpected impedance of highly unstructured environments using compliant materials and variable stiffness[6]. Among several artificial muscles, electro-active ionic soft actuators have many advantages, which include large bending deformations at low stimulating voltages <2 V, small power consumption, high energy density and self-sensing actuation[7–23]. These actuators deform due to ion migration in an polyelectrolyte membrane in response to electrical field applied to two sandwiched electrodes[5,24]. Hence, actuation performance is determined by the number and speed of moving ions, which are regulated by a combination of frictional force inside the polyelectrolyte and driving force of the electrical field. Therefore, many reports about electrode materials aim to increase the driving force and input energy by achieving high electrical conductivity and large capacitance of active electrodes[5,25–34]. And previous studies in electrolytes attempt to reduce the frictional force and energy loss via forming efficient pathway of ionic conducting channels inside ionic membranes[24,35–38]. However, polyelectrolytes have received much less attention than their electrode counterparts, possibly due to the challenge of generating a continuous conducting

[1]National Creative Research Initiative for Functionally Antagonistic Nano-Engineering, Department of Mechanical Engineering, Korea Advanced Institute of Science and Technology (KAIST), 291 Daehak-ro, Yuseong-gu, Daejeon 34141, Republic of Korea. [2]Department of Mechanical and Production Engineering, Aarhus University, Katrinebjergvej 89 G-F, 8200 Aarhus N, Denmark. [3]Transmission Electron Microscopy Laboratory, KAIST Analysis Center for Research Advancement, Korea Advanced Institute of Science and Technology (KAIST), 291 Daehak-ro, Yuseong-gu, Daejeon 34141, Republic of Korea. [4]Active Materials and Smart Living Laboratory, Department of Mechanical Engineering, University of Nevada, Las Vegas (UNLV), Las Vegas, NV 89154, USA. ✉e-mail: ikoh@kaist.ac.kr

network without significantly sacrificing the mechanical strength[24,35]. This challenge arises from the inherent distinction between two properties stemming from separate Nafion chains: ionic conductivity is derived from hydrophilic side chains, while mechanical strength mainly originates from hydrophobic non-conducting domains. Because these components are not compatible, they give rise to distinct phases. Consequently, the distribution and proportion of these phases exert both positive and negative influences on both mechanical strength and ionic conductivity[39–41]. Although several block-ionomers were reported with an expectation that their self-assembly could be a good solution[24,35], weak self-assembly force of block copolymers could not be suitable for governing their morphology of hundred-micrometer thick membranes needed in these practical applications[42]. Therefore, developing a polyelectrolyte having efficient ion pathway networks and good mechanical stiffness in multiscale is a challenging issue needed to be solved urgently.

This communication reports a procedure for producing polyelectrolyte membranes with continuous conducting phase as depicted in Fig. 1a–c. Accordingly, we used Nafion as an ionomer and found proper solvent systems for the formation of Nafion micelles, which were aggregated during the addition of ionic liquids and interconnected during casting. A functionally antagonistic solvent procedure considering dielectric constants and electrostatic equilibrium resulted in highly ion-conducting polyelectrolyte membranes, which can be used to greatly enhance actuation performances of electro-ionic soft actuators in view of bending deflection (Fig. 1d), response time, durability, and bandwidth. Additionally, the resulting electro-ionic soft actuator was applied to construct a crawling inchworm soft robot and a kinetic tensegrity system as shown in Fig. 1e.

## Results

### Material selection

For realizing our idea, we selected Nafion as an ionomer due to three following reasons[43]. First, Nafion molecules contain hydrophilic side chains of perfluorovinyl ether with sulfonate end groups hanging on the hydrophobic backbones of tetrafluoroethylene, which can provide ionic conducting path and mechanical strength, respectively (Fig. 1a). Second, it was proved from theoretical and experimental data that Nafion molecules exhibit different configurations in dispersions according to the dielectric constants of solvents[44–46]. Low dielectric constants promote dispersion of the hydrophobic backbones and higher values cause tight packs of the hydrophobic backbones[44]. Furthermore, dielectric constants can be adjusted by mixing solvents and altering temperature[47]. Therefore, we tuned those conditions to generate Nafion micelles with hydrophilic side-chains on the surfaces and hydrophobic backbones in the cores (Fig. 1a). Third, random distribution of the hydrophilic side chains cannot totally cover the hydrophobic backbone of Nafion, which could physically cross-link these micelles for mechanical reliable membranes as shown in Fig. 1c[48].

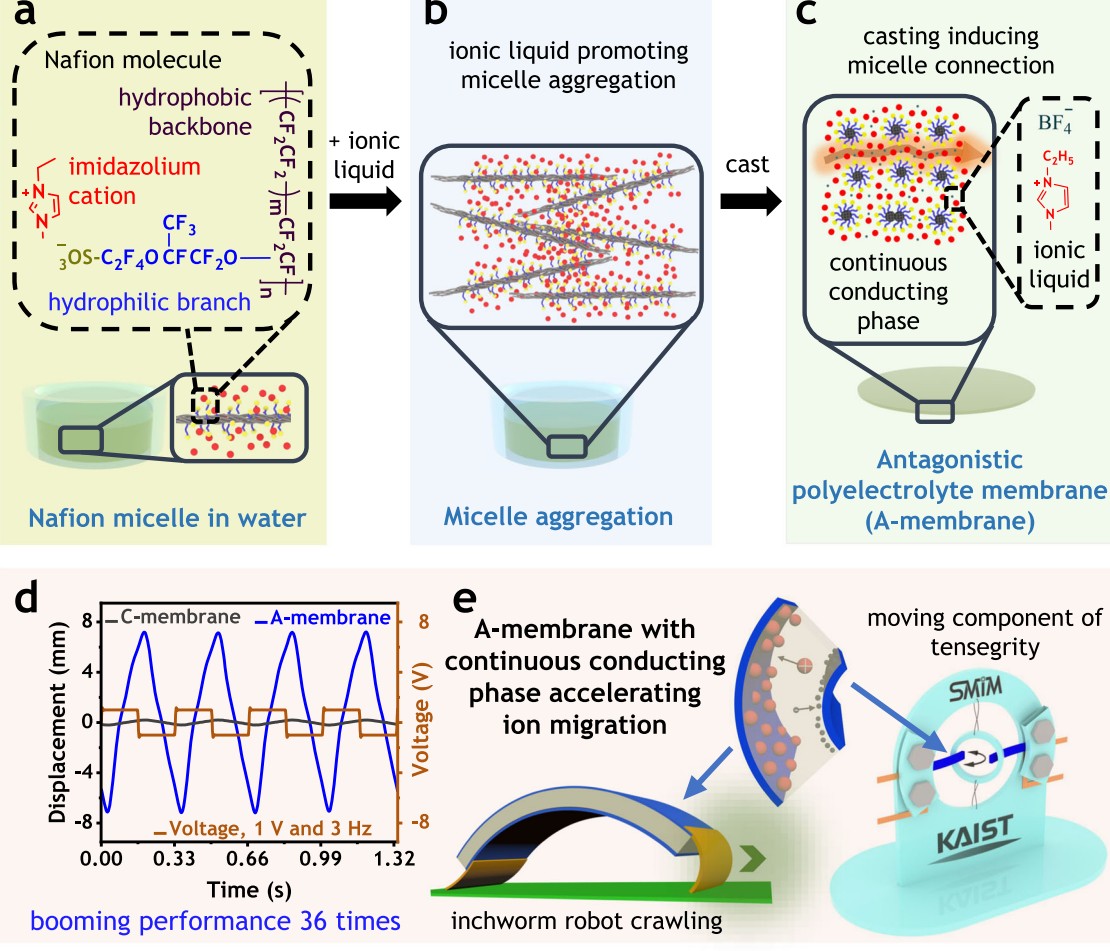

**Fig. 1 | Antagonistic polyelectrolyte membrane for ionic artificial muscle.** Procedure for preparing antagonistic polyelectrolyte membrane (A-membrane): (**a**) Nafion micelle in water, (**b**) Micelle aggregation, and (**c**) A-membrane. **d** Typical bending deformation. **e** Application of A-membrane to ionic actuators for use in an inchworm-mimetic soft robot and a dynamic tensegrity system.

The functionally antagonistic solvent procedure based on electrostatic equilibrium and hydrophilic/hydrophobic polymer chains was newly developed to make efficient polyelectrolyte membranes suitable for electro-ionic soft actuators. To form the aforementioned micelles and membranes, we chose deionized water (an excellent solvent for the side chains of Nafion, dielectric constant of about 80 at room temperature) and dimethylacetamide (DMAc, a good solvent for Nafion, dielectric constant of about 40 at room temperature)[47]. The former solvent causes micelle evolution (Fig. 1a) and the latter solvent aids micelle connection for membrane formation (Fig. 1c). To determine a proper ratio between these two solvents that meet the two requirements, we calculated Hansen solubility parameters (HSP) and used as a guideline for experiments (See Methods for detail calculation and listed values in Supplementary Tables 1 and 2)[44,45,49–51]. About 8.5% of DMAc in water was a suitable ratio for this purpose as shown in Supplementary Table 1. For complete dissociation of sulfonate groups and their strong electrostatic repulsion, which facilitates the molecular self-assembly, we replaced protons with 1-ethyl-3-methylimidazolium cations by neutralizing the ionomer with 1-ethyl-3-methylimidazolium hydroxide (Supplementary Fig. 1).

To develop dry-type ionic soft actuators operating in open air, ionic liquids are usually combined with ionomers. In this study, 1-ethyl-3-methylimidazolium tetrafluoroborate ionic liquid ($EMImBF_4$) was used because of its high ionic conductivity and big difference in molecular size between cation and anion[24]. Furthermore, because the presence of soluble salts in water reduces the repulsive Coulombic force among similar charged ions due to lowering dielectric constant, $EMImBF_4$ is selectively mixed with the hydrophilic side chains rather than the hydrophobic backbones[24,52]. Adding this ionic liquid weakened the repulsive force among the negative-charged surfaces of these micelles and made them aggregate, but did not affect their hydrophobic backbone cores (Fig. 1b). Casting these aggregated micelles into a free-standing polyelectrolyte film was applied at the final step (Fig. 1c).

## Micelle properties

For micelle formation, water of about 20.0 g was slowly added to Nafion solution of 0.7 g (Dupont, D2021, 20 wt%) over one hour at high temperature (90 °C). The dispersion was stirred overnight at the same speed and temperature before being left to cool down to room temperature and still for at least three days. Transmission electron microscopy (TEM) was used to investigate the morphology of these micelles. For good TEM visibility, this Nafion dispersion was neutralized by cesium hydroxide instead of 1-ethyl-3-methylimidazolium hydroxide[46]. To maintain the original structure of micelles, cryogenic TEM (cryo-TEM) was employed to take two-dimensional (2D) images because its wet-state samples preserved the structures of Nafion micelles in water as represented in Fig. 2a. Cryo-TEM tomography was also employed for detail examination because this technique enables to observe a position at different angles for reconstructing a three-dimensional (3D) image as shown in Fig. 2d.

All cryo-TEM images (Fig. 2b, c and Supplementary Fig. 2) and tomography results (Fig. 2e, Supplementary Fig. 3, and Supplementary Movie 1 and 2) evidently and consistently exhibit three crucial traits of the developed micelles, which are wormlike shape, extensive network, and high stability. As it can be clearly discerned from all TEM data, especially 2D and 3D TEM images in Fig. 2b, e and Supplementary Fig. 3, most micelles had wormlike shape existing in extensive network. These two characteristics are strongly supported by TEM images in Supplementary Figs. 2 and 3, which show that wormlike and networked micelles were completely dominant over a broad range of Nafion concentrations (Supplementary Fig. 2) and highly visible in different angles (Supplementary Fig. 3). Micelle stability can immediately be recognized by watching Fig. 2c, Supplementary Figs. 2 and 3, and Supplementary Movie 1 and 2, which display that these micelles did not

aggregate despite their dense and extensive networks at high concentrations. These TEM data agrees with the HSP calculation ("Methods" section and Supplementary Table 2) and previous studies[44–46].

These three micellar traits can be clarified by the Nafion-water intermolecular interaction and the random distribution of the hydrophilic side chains in Nafion molecules. First, at high temperature and dilute dispersion in high dielectric constant medium of water, strong repulsive force among sulfonate groups of a Nafion molecule could make this molecule to elongate. When temperature naturally decreases and the dispersion is left still, hydrophobic effect causes these elongated Nafion molecules to assemble into wormlike micelles. Second, because few long hydrophilic side chains (equivalent weight of 1000–1100) randomly distribute in long Nafion molecules (molecular weight of $10^5$–$10^6$ Da)[43], one Nafion molecule could simultaneously participate in forming some micelles, which are responsible for major formation of micellar networks. Finally, because water is selectively miscible with the hydrophilic side chains of Nafion rather than its hydrophobic backbones, Nafion molecules assemble into micelles whose hydrophilic surfaces cover the hydrophobic cores. In the high dielectric constant medium of water, the negative charges of sulfonate groups in the micellar surfaces strongly repulse each other, preventing these micelles from approaching close to each other. The electrostatic equilibrium explains for great stability of these functionally antagonistic micelles, which have the hydrophobic backbones in the cores and the hydrophilic side chains on the surfaces, without aggregation despite their high density.

As expected, adding $EMImBF_4$ to the as-prepared micellar dispersion caused aggregations as shown in Fig. 2f–j and Supplementary Fig. 4. Consequently, while a Nafion dispersion in water scattered light more in the presence of $EMImBF_4$, its counterpart in DMAc remained the same. The straight red scattered lines in Fig. 2f, h, i resulted from Rayleigh scattering and indicate that the sizes of particles in these dispersions were so small that Tyndall effect appears. By contrast, the asymmetrically scattered light in Fig. 2g was a result of Mie scattering and means that the sizes of particles in this dispersion were relatively big. Most importantly, Nafion molecules had formed micelles in water and aggregated while adding $EMImBF_4$, but dispersed well in DMAc regardless of the presence of $EMImBF_4$ ionic liquid. This inference is confirmed by dynamic light scattering (DLS) and zeta potential data presented in Fig. 2j and Supplementary Fig. 4. Accordingly, the addition of $EMImBF_4$ induced significant increases in both particle size and zeta potential of micelles in water (Fig. 2j and Supplementary Fig. 4a, b), but slight change for Nafion dispersion in DMAc (Fig. 2j and Supplementary Fig. 4c, d). These particle size and zeta potential data are inter-supported because only particles with zeta potential lower than −30 mV (the dash line in Fig. 2j) have strong electrostatic repulsions enough to make them stable[53]. In addition, negative values of zeta potentials validate that these micelles had negative charges of the sulfonate groups of Nafion on the surfaces.

## Membrane properties

The presence of water had an impact on both micellar rheology (as shown in Supplementary Fig. 5) and the ultimate membrane structure. The recipe provided in Supplementary Table 1 represents an optimized condition that took into account numerous parameters throughout the entire process, from preparing the Nafion solution to casting the membrane. Notably, during the casting step, the arrangement of Nafion molecules within the micelles was maintained due to the casting temperature being significantly lower (90 °C) than the glass transition temperature of Nafion[54,55]. Therefore, casting these as-made networked micelles with ionic hydrophilic surfaces covering hydrophobic cores can form the desired polyelectrolyte membranes and continuous conducting matrix of the hydrophilic side chains plasticized by $EMImBF_4$ greatly improves ionic conductivity. And nonconducting domains of hydrophobic backbone stabilized by physical

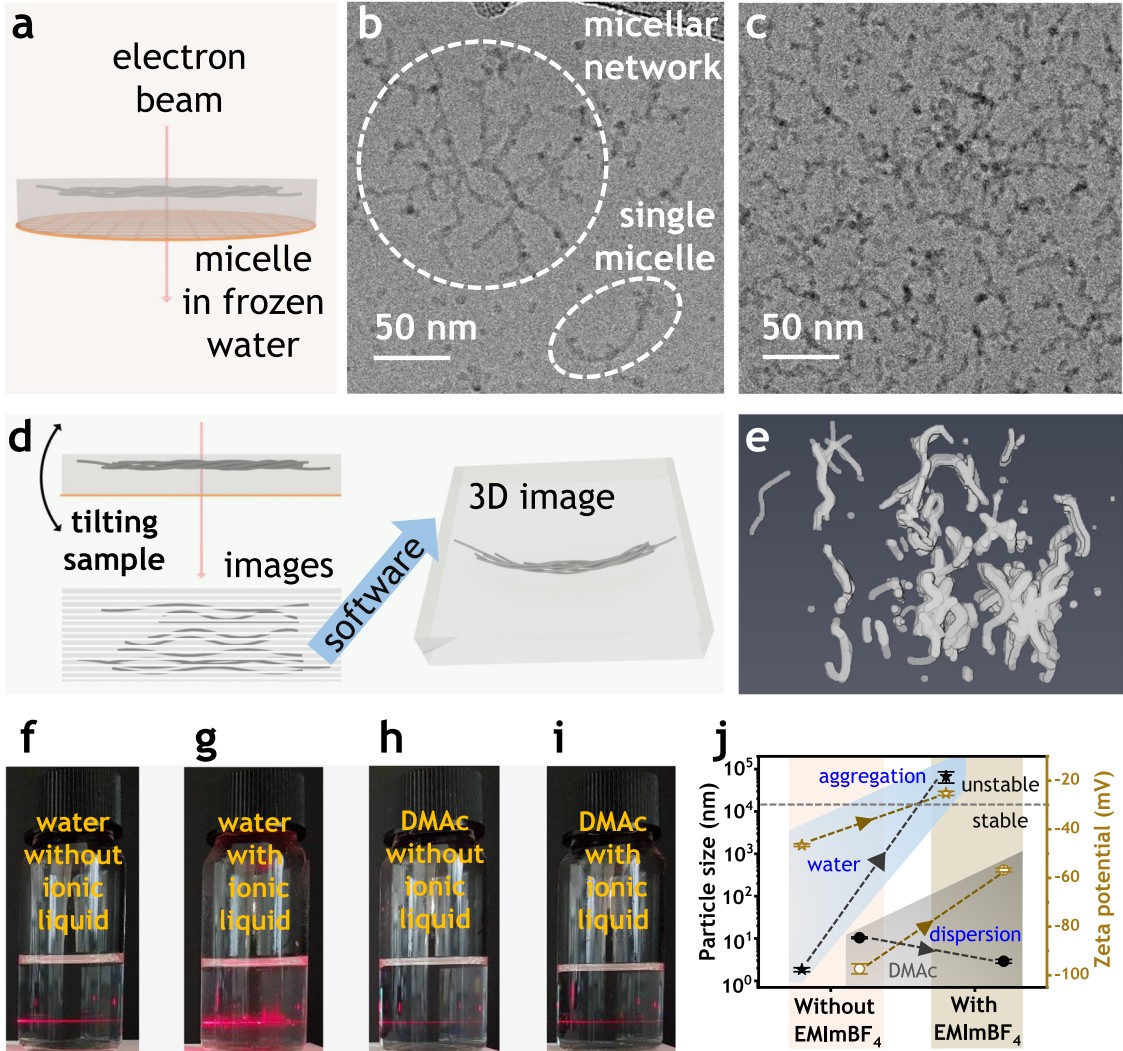

**Fig. 2 | Properties of micelle. a** Cryogenic transmission electron microscopy (Cryo-TEM) principle. **b** Cryo-TEM image with clear micellar network (in cycle) and single micelle (in ellipse) at low concentration. **c** Dense micelles at high concentration. **d** Cryo-TEM tomography principle. **e** 3D image reconstructed from cryo-TEM tomography. **f–i** Laser scattering of Nafion dispersions in water and dimethylace-tamide (DMAc). **j** Micelle aggregation due to the presence of EMImBF4 ionic liquid according to particle size and zeta potential results.

cross-linkers enhance mechanical strength of the membrane. The whole procedure for fabricating polyelectrolyte membranes is presented in Supplementary Fig. 6. The measured ionic conductivity, specific capacitance, and mechanical stiffness of ionic exchangeable membranes are summarized in Fig. 3 and Supplementary Figs. 7–9.

Figures 3a, b demonstrate a vital role of DMAc in the formation of polyelectrolyte membranes. Only using pure water generated small fragments as shown in Fig. 3a, because water and EMImBF$_4$ are such poor solvents for the hydrophobic backbones of Nafion that no inter-bonding among these micelles occurred. By contrast, adding 8.5% DMAc into water produced free-standing membranes as shown in Fig. 3b, because DMAc is a good solvent for Nafion. During casting, water evaporated much faster than DMAc due to 0.6-time lower boiling point, which enriched DMAc portion and promoted such stable physical cross-linkers binding all micelles in a mechanically strong continuous film as shown in Fig. 3b.

The electrochemical impedance spectroscopy (EIS) and ionic conductivity were measured to investigate the basic properties of the polyelectrolyte membranes using the experimental setup shown in Fig. 3c. The EIS results of pure polyelectrolyte membranes are shown

in Fig. 3d and Supplementary Fig. 7a–c were used to calculate ionic conductivities as presented in Fig. 3e. Comparing to the conventional membrane (C-membrane), the antagonistic membrane (A-membrane) had three times higher ionic conductivities in the whole tested temperatures ranging from 24 to 65 °C. For the cyclic voltammetry (CV) test (the measurement setup in Fig. 3c), both surfaces of membranes were coated by poly(3,4-ethylenediox-ythiophene)-poly(styrenesulfonic acid) (PEDOT:PSS) electrodes to evaluate capacitance characteristics in the actuator level. The CV results of the actuators shown in Fig. 3f and Supplementary Fig. 8a, b were used to determine specific capacitance as summarized in Fig. 3g. Consequently, despite the same electrode system, A-membrane resulted in much higher specific capacitance than C-membrane. These values increased from 1.7 to 4.5 times when excitation frequencies ranged from 0.1 to 5.0 Hz. These enhance-ments in specific capacitances mean that, with the same electrodes and testing conditions, A-membrane had more ions going to elec-trodes than C-membrane, especially at high frequencies. This could be due to two following reasons. First, A-membrane with higher ionic conductivity should have supplied more ions to neutralize more

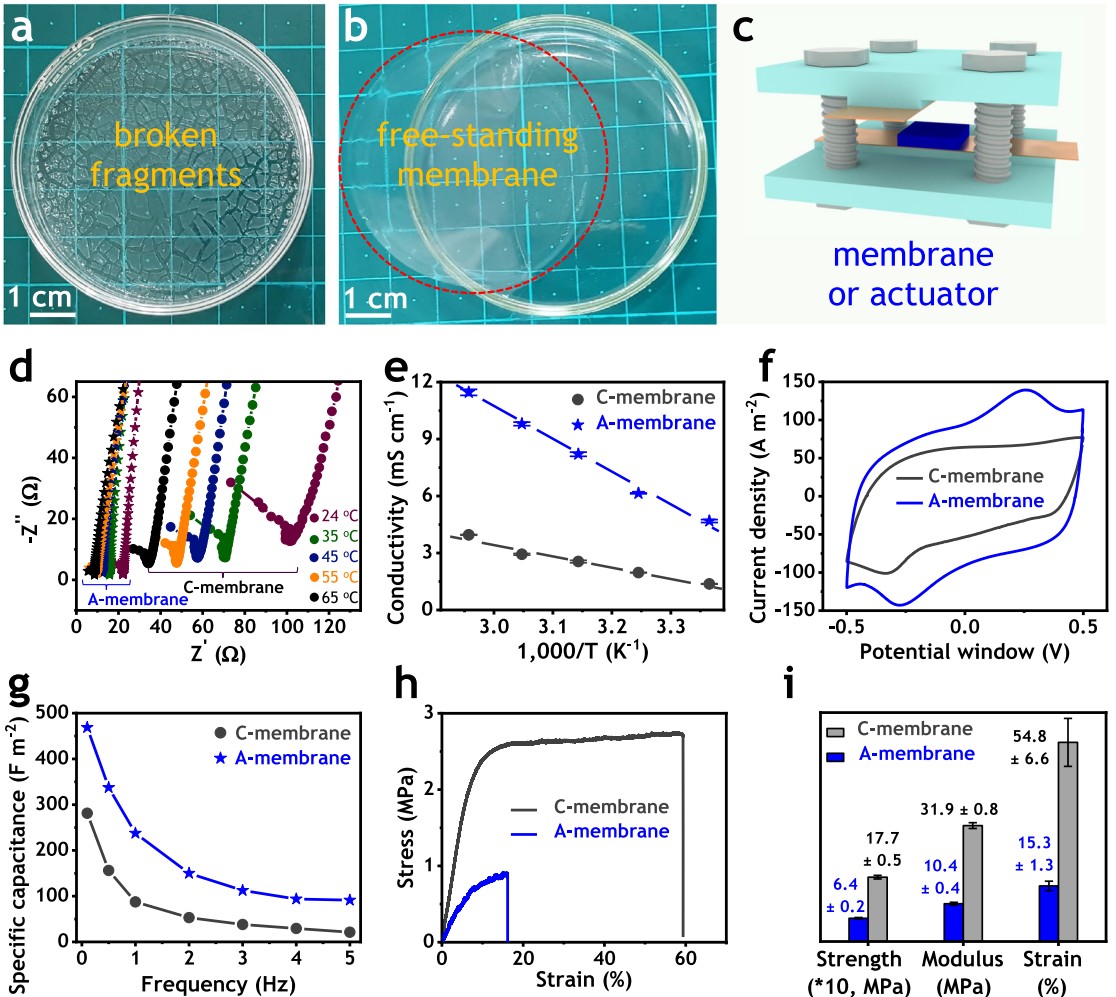

**Fig. 3 | Properties of A-membrane. a** 0% and (**b**) 8.5% of DMAc in water affecting on the formation of membrane. **c** Measurement setup for electrochemical impedance spectroscopy (EIS) of electrolyte membranes and cyclic voltammetry (CV) of ionic actuators. **d** Typical EIS spectra. **e** Ionic conductivity. **f** Typical CV curves. **g** Specific capacitance. **h** Typical tensile curves. **i** Mechanical strength.

charges in electrodes than C-membrane. Therefore, the improvement of specific capacitance and ionic conductivity are inter-supported and originated from more efficient continuous conducting phase in the A-membrane. Second, A-membrane with stronger bonding to PEDOT:PSS electrodes should have facilitated more ion movement between electrolyte and electrodes than C-membrane because the surfaces of the former membrane made of Nafion micelles in water had much more sulfonate groups than those of the latter membrane made of Nafion dispersion in DMAc.

The detailed geometry of the polyelectrolyte specimen for the tensile test are shown in Supplementary Fig. 9a. Tensile curves shown in Fig. 3h and Supplementary Fig. 9b were utilized to compute mechanical stiffness and strength as disclosed in Fig. 3i. Thereupon, A-membrane had lower Young modulus and yield strength than C-membrane. This proves the effect of phase distribution on membrane properties and suggests that A-membrane had less connected non-conducting domains, which majorly contributed to relatively lower mechanical stiffness than C-membrane. The tensile test results of A- and C-membrane actuators are shown in Supplementary Fig. 10. All data of these three characterizations indicate that A-membrane had continuous conducting phases for high ionic conductivity and well-connected non-conducting domains for compromised mechanical stiffness, which are basically required for high-performance ionic soft actuators.

## Actuation performance

The A-membranes were used for fabricating electro-ionic soft actuators according to a procedure illustrated in Supplementary Fig. 11. In this study, we focused on the polyelectrolyte membrane itself and only used PEDOT:PSS as a simplest conducting electrode without any other fillers and additives. Actuation performance was investigated by using the experimental setups shown in Fig. 4a and Supplementary Fig. 12. The cross-sectional scanning electron microscopy (SEM) image in Fig. 4b clearly shows that the A-membrane actuator had a symmetrically sandwiched structure with total thickness of about 100 μm. A typical harmonic response of the actuator in Supplementary Movie 3 displays a large tip-to-tip bending displacement close to 35 mm at a low voltage of 1.0 V and applied frequency of 0.1 Hz. The developed electro-ionic soft actuators were intensively investigated by measuring displacements, durability, and blocking forces over broad excitation ranges of direct current (DC) and alternative current (AC) voltages. The obtained results were collected in Fig. 4c–i and Supplementary Figs. 13–16.

Figure 4c, d and Supplementary Fig. 13 clearly present significant improvement in DC response. According to Fig. 4c, the A-membrane actuator generated over 1.5-time higher displacement and 10-time faster response than the C-membrane actuator at the step input of 0.3 V. This trend is consistent as the applied voltage varies from 0.1 to 0.3 V (Supplementary Fig. 13a). Especially, the A-membrane actuator

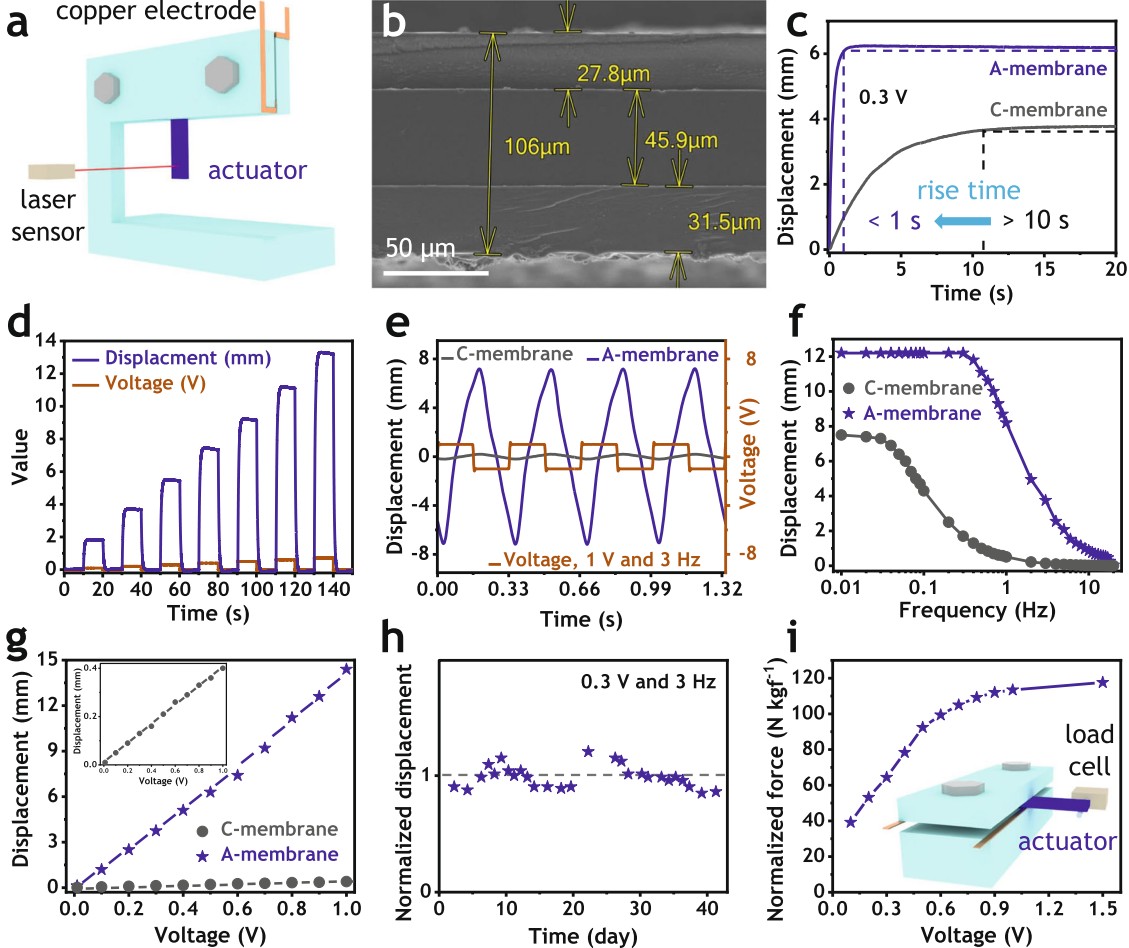

**Fig. 4 | Actuation performance. a** Measurement setup for displacement. **b** Cross-sectional scanning electron microscopy (SEM) image of actuator made of A-membrane. **c** Rise time at 0.3 V direct current (DC). **d** Step DC responses from 0.1 to 0.7 V of actuator made of A-membrane. **e** Typical bending displacement at 1 V and 3 Hz. Displacements according to (**f**) Frequencies at 0.3 V and (**g**) Voltages at 3 Hz. **h** Durability of actuator made of A-membrane (over 42 days, 11 million cycles). **i** Blocking force of actuator made of A-membrane.

resulted in no back-relaxation over an extensive period of time for 35 minutes (Supplementary Fig. 13b). Maintaining the step response for extremely long time without performance degradation is an exceptional result that has not been achieved in electro-ionic soft actuators. The fast response without back-relaxation is further confirmed when stimulating voltages of square shapes were changed in a short interval of 10 seconds (Fig. 4d). As can be seen in this figure, when voltages switched from 0.1 to 0.7 V, this actuator rapidly reached to maximum deflections, which maintained without any displacement drop in the on stages, and immediately recovered its original positions in the off stages; bending displacements linearly increased with voltage amplitudes. These fast and large responses without back-relaxation could greatly enhance repeatability and accuracy for soft robotic devices made up of the electro-ionic soft actuators.

In order to further examine the A-membrane actuators, wide ranges of AC voltages with broad frequency bandwidth were applied. A typical response at 1 V and 3 Hz in Fig. 4e shows a huge improvement of over 35 times in the tip bending deflection. This advancement in bending deflection continued in the broad excitation frequency range from 0.01 to 20.0 Hz in Fig. 4f and in the applied voltage from 0.01 to 1.00 V in Fig. 4g. Most interestingly, Fig. 4f shows that the A-membrane actuator maintained tip displacement from 0.01 Hz to 0.30 Hz, while the C-membrane actuator shows a steep drop of the tip displacement as the excitation frequency increases. This suggests that all dissociated ions of ionic liquid in the A-membrane under electrical fields can move

much faster to counter electrodes due to efficient ion pathway of the ionic conducting network, resulting in much broader bandwidth. Because of 10-time reduction in rise time, the A-membrane actuator maintained maximum displacement up to 0.30 Hz and performed visible bending up to 20 Hz while the C-membrane actuator started to reduce displacement at 10-time lower frequency of 0.03 Hz and showed very small bending deflection at 2 Hz (Fig. 4f and Supplementary Fig. 14). These actuators were further tested at a high frequency of 3 Hz and different voltages ranging from 0.01 to 1.00 V. The results in Fig. 4g and Supplementary Fig. 15 prominently display that the A-membrane actuator kept performing larger bending displacement of 15 mm at 1 V, while the C-membrane actuator presented minor displacement of merely 0.4 mm at the same voltage. It is worth mentioning that the A-membrane actuator with much larger bending deflection at higher frequency up to 20 Hz could broaden their practical applications in real fields.

The A-membrane actuators were further examined in view of durability and blocking force because these two properties are very important for several applications. For durability, the A-membrane actuator was continuously activated with a harmonic excitation of 0.3 V and 3.0 Hz over 40 days. The displacement values were normalized with the averaged displacement over 40 days. As presented in Fig. 4h and Supplementary Figs. 16 and 17, the A-membrane actuator exhibited excellent cyclic stability over 42 days, indicating about 11 million cycles in the continuous harmonic excitation. The durability of

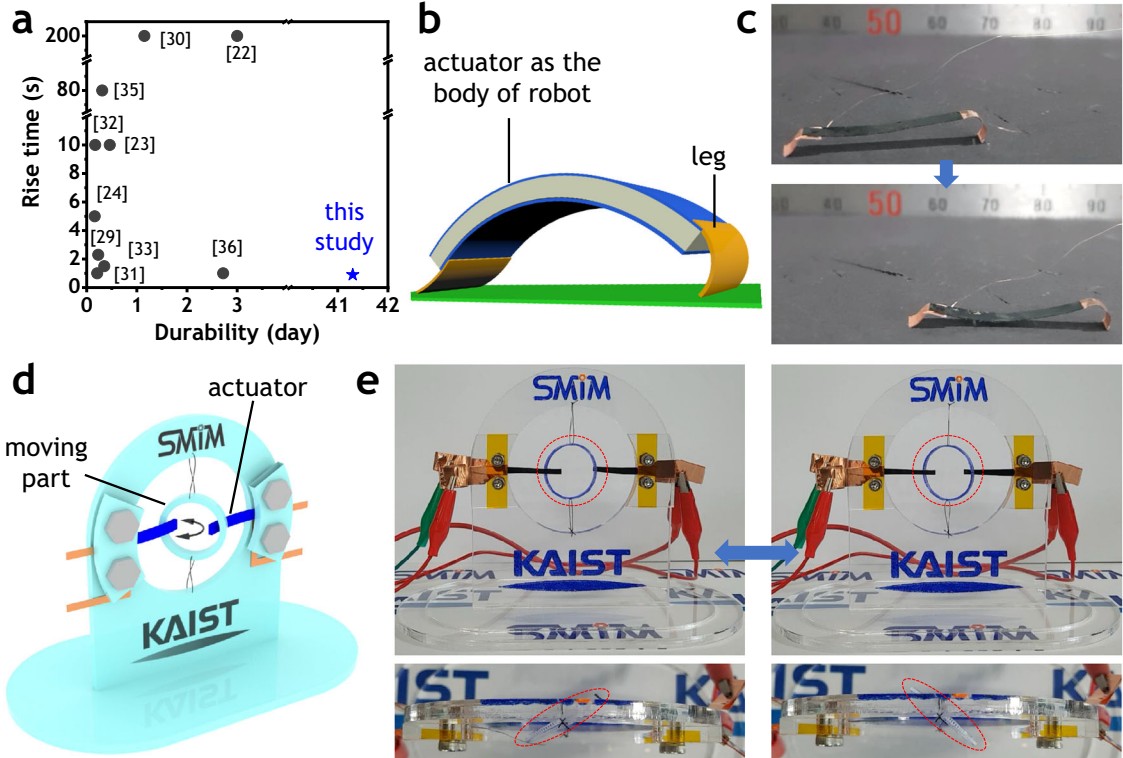

**Fig. 5 | Actuation comparison and demonstration. a** Comparison of rise time and durability of the developed actuator to some literature data. Demonstrations of (**b**) an inchworm-mimetic soft robot and (**c**) Locomotion of inchworm robot. **d** A dynamic tensegrity system made of the developed actuator. **e** Dynamic motion of simple tensegrity system under electrical stimulus.

the C-membrane actuator under continuous sinusoidal excitation with peak voltage of 1.5 V and the frequency of 3.0 Hz is relatively poor as shown in Supplementary Fig. 18. There are two reasons for this exceptional durability. The first reason is that large amount of sulfonate ionic groups on the both surfaces of A-membrane and PEDOT:PSS made strong bonding between polyelectrolyte and electrode layers without delamination. By contrast, general C-membrane formed poor bond with PEDOT:PSS because of its hydrophobic surfaces made up of tetrafluoroethylene main chains. The second reason is the efficient ionic nanochannel networks of the polyelectrolyte. In case of the C-membrane with relatively poor ionic conductivity, ionic liquid can slowly accumulate to the interface between electrode and polyelectrolyte during continuous operation in the durability test. This accumulated ionic liquid gradually separates electrode and polyelectrolyte and eventually leads to physical delamination, which is also accelerated by repeated bending deformations. However, because cations and anions in the A-membrane can move much faster, the formation of the accumulated ionic liquid layer is suppressed and the strong bond at the interface of the electrode and the electrolyte even after long-term excitation is maintained without delamination. Slight fluctuation of durability results shown in Fig. 4h originates from temperature and humidity changes in ambient air. The A-membrane actuator also had remarkable blocking force as presented in Fig. 4i. Accordingly, the normalized blocking force with actuator weight increases from 40 to 110, when applied voltages varied from 0.1 to 1.0 V. Despite only using PEDOT:PSS, the A-membrane actuator generated much stronger blocking force than recently reported actuators made up of advanced electrode materials including covalent organic framework, graphene and molybdenum sulfide[28,56].

It is worth mentioning that all actuation performances are greatly enhanced including short rise time, no back-relaxation, large bending deformation at low voltages and broad bandwidth, long-term stability and high blocking force. The high qualities of the A-membrane could stand out when observing a comparison shown in Fig. 5a and Supplementary Table 3. According to Fig. 5a, while many reported actuators had long rise time up to 200 seconds and poor durability less than 3 days[22–24,29–33,35,36], the A-membrane actuator resulted in the smallest rise time less than one second and exceptional durability over 42 days under continuous harmonic excitation.

In order to prove the ability of practical applications, the developed actuator was used to construct a soft robot inspired by inchworm as shown in Fig. 5b, c and Supplementary Movie 4, and to excite a dynamic tensegrity system as presented in Fig. 5d, e and Supplementary Movie 5. The inchworm soft robot definitely crawled like a natural inchworm over a long distance and the dynamic component of the tensegrity system was clearly oscillated according to the excitation of the as-prepared actuators. These two successful demonstrations indicate that the A-membrane actuator has a lot of potential to be applied to several further applications.

## Discussion
In this study, we report a procedure to fabricate an efficient polyelectrolyte membrane needed for high-performance electro-ionic soft actuators. Considering hydrophilic–hydrophobic domains of Nafion and ionic liquid-involved electrostatic equilibrium, our functionally antagonistic solvent procedure resulted in micelles with ionic surfaces covering non-conducting cores. These micelles can form a continuous conducting phase in polyelectrolyte membranes suitable for high ionic conductivity and well-connected non-conducting domains for adequate mechanical strength. The A-membrane actuator shows much faster step response with an exceptionally short rise time <1 s, 36-time higher tip-to-tip bending displacement at the applied voltage of 1 V, remarkable long-term stability over 42 days, and 110 times normalized blocking force. The as-developed actuators surpassed actuation

perfomances of many other actuators reported by today and were successfully applied to develop a crawling inchworm robot and a dynamic tensegrity system. This study contribute to advances in polyelectrolyte membranes, ionic artificial muscle, and soft robotics as well.

## Methods

### Materials

Some main materials used in this research are Nafion solution (DuPont, D2021 contained about 20 wt% polymer, 34 wt% water, and 46 wt% 1-propanol), 1-ethyl-3-methylimidazolium tetrafluoroborate (IOLITEC, EMImBF$_4$, 98%), 1-ethyl-3-methylimidazolium chloride (IOLITEC, EMImCl, 98%), *N,N*-dimethylacetamide (Sigma-Aldrich, DMAc, anhydrous, 99.8%), dimethylsulfoxide (Sigma-Aldrich, DMSO, anhydrous, 99.9%), ion-exchange resin (Sigma-Aldrich, Amberlite IRN78 OH hydroxide form), poly(3,4-ethylenedioxythiophene)-poly(-styrenesulfonic acid) (Heraeus, PEDOT:PSS, CLEVIOS PH 1000), and milli-Q type 1 ultrapure water. 1-ethyl-3-methylimidazolium hydroxide (EMImOH) solution was prepared by passing EMImCl solution through a column filled with the ion-exchange resin (Supplementary Fig. 1).

### Calculation of solubility parameters for formulating solvent system

The conformation of polymer molecules in solutions could be predicted by Hansen solubility parameter (HSP), because it depends on the interaction between polymer and solvent molecules[49]. In general, good solvents have strong interactions with polymer molecules, which induce polymer chains to expand in open-coil conformation for good solutions. In contrast, poor solvents have weak interactions with polymer molecules, which cause polymer molecules to shrink in compact-coil structure, making poor dispersions or even precipitation. Especially, amphiphilic polymers can form micelles whose cores and shells are made of polymeric components that have weak and strong interactions with solvents, respectively.

We hypothesize that proper solvent systems interacting strongly with sulfonate groups in the side chains of Nafion, but weakly with tetrafluoroethylene backbone chains of this ionomer could cause Nafion molecules in solution to form micelles with sulfonate groups on the surfaces. Water was selected to be a main solvent in this research for the formation of Nafion micelles because it is miscible with the hydrophilic side chains of Nafion, but immiscible with the hydrophobic tetrafluoroethylene backbones of this ionomer. A small portion of DMAc, which is considered a good solvent for Nafion, was added to adjust solubility favorable to micellar formation and membrane casting. Micelles were evolved from Nafion D2021 solution, which contains 1-propanol. A small amount of EMImBF$_4$ was also added as the ion source aiding actuator application. Therefore, the solvent system for Nafion micelles contains major water and small percentages of DMAc, 1-propanol, and EMImBF$_4$.

In order to find proper ratios among these components for polyelectrolyte membranes, we first estimated HSP of solvent mixtures ($\delta_{mj}$) from volume fraction ($\Phi_i$) and solubility ($\delta_i$) of each solvent via Eq. (1)[49].

$$\delta_{mj} = \sum \Phi_i \delta_i \tag{1}$$

Then, we calculated distance parameter ($R_a$) of solvent mixture with respect to Nafion according to Eq. (2) and used this value to predict the morphology of Nafion molecules[49]. Normally, a solvent system can dissolve a polymer if its $R_a$ is smaller than the interaction radius ($R_O$) of that polymer. Because $R_O$ of Nafion is not available, we compare $R_a$ values of our solvent systems to those values of N-methyl-2-pyrrolidone (NMP) and water/2-propanol (1:1 by volume), which were proved to be good and poor solvents for Nafion, respectively[45].

$$Ra^2 = 4(\delta_{d-solvent} - \delta_{d-polymer})^2 + (\delta_{p-solvent} - \delta_{p-polymer})^2 + (\delta_{h-solvent} - \delta_{h-polymer})^2 \tag{2}$$

We used HSP and $R_a$ as guidelines to adjust the ratio of those solvent components in order to produce good electrolyte membranes. Several Nafion dispersions were tested and finally a proper solvent mixture was selected for the A-membrane (Supplementary Table 1). A conventional solvent system, named C-membrane, that contains major DMAc and a small amount of EMImBF$_4$, was also provided in Supplementary Table 1. HSP and $R_a$ values of A-membrane, C-membrane, and related solvents were summarized in Supplementary Table 2. According to Supplementary Table 2, our developed mixture could be a poor solvent system for Nafion because its $R_a$ value (29.72 MPa$^{1/2}$) is much higher than that of water/2-propanol (20.09 MPa$^{1/2}$), which was proved to be a poor solvent mixture and induced Nafion molecules to form compact-coil conformation[45]. Therefore, the A-membrane resulted in Nafion micellar dispersions. In contrast, the C-membrane is an excellent solvent system for Nafion, because its $R_a$ value (1.66 MPa$^{1/2}$) is much smaller than that of NMP (2.69 MPa$^{1/2}$), which was proved to be a good solvent and made Nafion molecules to expand in open-coil conformation[45].

### Preparation and characterizations of Nafion micelles

First, we checked micellar formation by cryogenic transmission electron microscopy (cryo-TEM). Nafion Micelles were prepared by diluting Nafion D2021 solution. On a hot plate at 90 °C and 500 rpm magnetic stirring speed, 20.00 g of water was slowly added to 0.70 g of Nafion solution over 1 h and the resulted mixture was stirred over night before being left to cool down to room temperature and still for at least three days. After that, 1 ml of this dispersion was neutralized by cesium hydroxide solution. The sample was deposited on a TEM grid by using a vitrobot and characterized by a cryo field emission TEM (200 kV, Glacios, Thermo Fisher Scientific).

Second, we checked static laser light scattering, dynamic light scattering (DLS), and zeta potential. The previous dispersion for TEM analysis became too viscous for DLS test after adding EMImBF$_4$ ionic liquid. Therefore, we made a diluter dispersion. Accordingly, on a hot plate at 90 °C and 500 rpm magnetic stirring speed, 0.75 g of DMAc was slowly added to 0.70 g of Nafion solution over 15 min followed by the addition of 48.00 g of water over one hour. The resulting mixture was stirred over night before being left to cool down to room temperature and still for at least three days. On the one hand, on a hot plate at 90 °C and 500 rpm magnetic stirring speed, this dispersion was diluted with a mixture of 0.75 g of DMAc and 5.75 g of water to be the sample without ionic liquid. On the other hand, on a hot plate at 90 °C and 500 rpm magnetic stirring speed, this dispersion was neutralized by 2.50 g of EMImOH solution in water before being added 0.20 g of EMImBF$_4$ in 0.75 g of DMAc and 3.05 g of water to be sample with ionic liquid. Two samples in DMAc were prepared in the same procedures except for replacing water by DMAc. These four samples were checked with static laser light scattering, DLS, and zeta potential.

### Casting and characterizations of polyelectrolyte membranes

A typical procedure for casting polyelectrolyte membranes is described as follows. On a hot plate at 90 °C and 500 rpm magnetic stirring speed, 0.75 g of DMAc was slowly added to 0.70 g of Nafion solution over fifteen minutes followed by the addition of 12.05 g water over 1 h. The resulted mixture was stirred over night before being left to cool down to room temperature and still for at least 3 days. After that, on a hot plate at 90 °C and 500 rpm magnetic stirring speed, this dispersion was neutralized by 2.50 g of EMImOH solution in water before being added 0.22 g of EMImBF$_4$ in 0.75 g of DMAc and 3.03 g of water. Finally,

after stirring for three hours, this dispersion was cast on a glass petri dish in an oven at 90 °C for two days before applying vacuum at the same temperature for at least three days, resulting in A-membrane with the thickness of ~80 μm (Supplementary Table 1 and Supplementary Fig. 6a). C-membrane was prepared as follows. 0.20 g of net Nafion, which was dried from Nafion D2021 solution in an oven at 60 °C for one day, was stirred in 15 g of DMAc at 80 °C for three days. The obtained dispersion was then mixed with 0.12 g of EMImBF₄ in 5 g of DMAc at the same temperature for at least 1 day. Finally, this dispersion was cast on a glass petri dish in an oven, resulting in C-membrane with the thickness of ~80 μm (Supplementary Table 1 and Supplementary Fig. 6b). These membranes were used for fabricating actuators.

Membranes that were prepared according to these previous procedures and had three-time thicker were used for electrochemical impedance spectroscopy (EIS, frequency ranging from 2.5 MHz to $10^3$ Hz at potential amplitude of 0.1 V) and tensile tests (Supplementary Fig. 9)[57].

### Fabrication and characterization of actuators

The procedure for fabricating actuators is illustrated in Supplementary Fig. 11. First, 10 g of PEDOT:PSS dispersion was doped by adding slowly 0.5 ml DMSO and stirred for 1 day at room temperature. Then, a Nafion membrane was fixed on a glass plate by polyimide tapes before being cast by the doped PEDOT:PSS dispersion. The electrode dispersion was dried on a hot plate at 90 °C for 90 minutes. After that, the doped PEDOT:PSS dispersion was cast on the other side of the electrolyte membrane before drying a hot plate at 90 °C for 90 min. Finally, the obtained sandwich structure was further dried in vacuum at 70 °C for two days, resulting in an actuator ready for cyclic voltammetry (CV) and actuation performance tests[57].

Actuators with free length of 2 cm were activated by direct current (DC) and alternative current (AC) voltages. Bending displacement and blocking force were measured by a laser sensor (LK-H080, KEYENCE) and a load cell (LVS-5GA, KYOWA), respectively (Supplementary Fig. 12). During checking actuators, the temperature in the testing room was maintained between 18 and 20 °C and the humidity was between 60 and 90%.

## Data availability

All data needed to evaluate the conclusions in the paper are present in the paper and/or the Supplementary Information. Additional data are available from the corresponding author upon request.

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

## Acknowledgements

This work was supported by Creative Research Initiative Program (2015R1A3A2028975) funded by National Research Foundation of Korea (NRF). This work was also supported by the National Research Foundation of Korea (NRF) grant funded by the Korea government (MSIT) (RS-2023-00302525).

## Author contributions

Conceptualization: V.H.N. and I.-K.O.; Investigation: V.H.N., I.-K.O., and K.J.K.; Methodology: V.H.N.; Resources: S.O.; Data curation: V.H.N., S.O., R.T., H.Y., S.-G.L., and M.G.; Visualization: V.H.N. and M.M; Writing – original draft: V.H.N.; Writing – review & editing: V.H.N., S.O., M.M., K.J.K., and I.-K.O.; Supervision, Funding acquisition and Project administration: I.-K.O.

## Competing interests

The authors declare no competing interests.
