## [Peer Review File · Nature Communications]

REVIEWER COMMENTS

Reviewer #1 (Remarks to the Author):

The paper reported a newly hemostasis-inspired membrane with an ionic surface enclosing non-conducting cores. The synthesis idea for Nafion micelle is interesting. Compared to the conventional membrane, the new structured membrane displayed three times higher ionic conductivities. In this paper, the new polyelectrolyte membrane is used for an ionic actuator showing good performance in terms of response amplitude, speed, and durability. I think it is a useful strategy for high-performance ionic actuator areas by using a new structured polyelectrolyte. Before publication, I would like to suggest the authors add more evidence, discussion, and comparison about important details of the new membrane such as the pore size and size distribution, is there any big change compared with traditional nafion? The pore structure would be acting as effective ion channels for ions transportation and deformation caused by ions immigration. In addition, I would like to suggest the authors redesign and replot their bionic synthetic idea about the hemostasis-inspired membrane in Fig1 (a-d), in order to let readers catch the idea quickly and easily.

Reviewer #2 (Remarks to the Author):

In this manuscript, the authors reported a hemostasis-inspired strategy to synthesize efficient polyelectrolyte membranes that have continuous conducting network suitable for electro-ionic actuator. And the developed artificial muscle shows great durability over 40 days under continuous actuation and much broader bandwidth below 10 Hz, which was applied to demonstrate an inchworm-mimetic soft robot and a kinetic tensegrity system. Overall, this manuscript was suitable to be accepted in Nature Communications after consideration of the following points:

1. The proportion of the dispersed system was determined by the calculation of R_a , but the experiments are lacking to verify whether the continuous conduction networks can also be formed at other proportions without sacrificing the mechanical strength.
2. The effects of Nafion colloidal concentration on the mechanical strength and the conductivity of the as-prepared film are not supported by the existing experimental data.
3. The controllability of the as-prepared films must be strengthened by more experimental results.
4. The mechanical properties of various films was displayed in Figure 3h-i, however, the yield strength is more important for the actuator than the tensile strength during the driving process, the data analysis method is suggested to adjust.
5. In addition to computational analysis, other experimental characterizations like FTIR and Raman can be also demonstrated to further prove the intermolecular effects of water and DMAc on the dispersion results.
6. The analysis about Figure 4f was mistakenly titled as Figure 4g in the manuscript.

Reviewer #3 (Remarks to the Author):

In this study, the authors present the fabrication and characterization of a hemostasis-inspired IMPC actuator. The concept of a bio-inspired chemistry approach is fairly novel, and the IMPC does show a significantly faster response speed. However, the mechanical properties of the actuator do not seem to be better compared to existing literature, and the reviewer would like to request the authors to provide an elaborate discussion on this particular subject matter. The following comments should also be addressed:

- Line 43: It is suggested to use consistent (10-times) terminology.
- Lines 70-77: The authors argue that the mechanical properties of the conductive network may decrease with conductive network improvement. It is suggested that the authors provide specific values for the "good" or "bad" mechanical stiffness. The reviewer suggests summarizing the literature results into a table.
- Figure 1 is very busy, most likely due to the embedded words, and it is suggested that the figure be further refined.
- Figure 2f-i caption: It is suggested that the authors elaborate on this particular caption with a discussion on the scattering light. The current caption does not explain the significance of the images.
- Figure 3c: It is suggested that the proper labels are included, similar to the following figures.
- Please explain why the conventional (C-membrane) shows a higher temperature dependence, as shown in Figure 3d.
- Lines 242-261: The authors mentioned that PEDOT:PSS was coated onto the membrane as an electrode. Please comment on the adhesion of nafion/PEDOT:PSS between the C- and H-membrane.
- Line 256: "H-membrane had a lower Young's modulus and tensile strength than C-membrane. This suggests that the H-membrane had fewer connected non-conducting domains." The reviewer has a minor concern about the above statement; the lower mechanical properties should not be directly correlated to a better connected network.
- Line 260: "In other words, normally PEDOT:PSS electrodes are much stiffer than the polyelectrolyte core layer." Please provide proper references and specific values.
- Line 263: "well-connected non-conducting domains for ideal mechanical stiffness." Following the previous comment, the reviewer would like to raise the question of the authors justifying the "ideal mechanical stiffness" as the H-membrane has much worse mechanical properties, both in strength and strain. Also, it is suggested that the authors provide mechanical testing results on both PEDOT:PSS coated and non-coated samples for comparison.
- Line 265: "The mechanical stiffness of the polyelectrolyte can be tailored by changing the ratio between water and DMAc, based on the functionally antagonistic solvent procedure." This statement should be removed unless the authors provide experimental results.
- Figures 4h & i: Is it possible to also add the long-term performance of the C-membrane to these graphs for comparison?

Authors' Responses to Reviewer #1

Reviewer's overall statement: *The paper reported a newly hemostasis-inspired membrane with an ionic surface enclosing non-conducting cores. The synthesis idea for Nafion micelle is interesting. Compared to the conventional membrane, the new structured membrane displayed three times higher ionic conductivities. In this paper, the new polyelectrolyte membrane is used for an ionic actuator showing good performance in terms of response amplitude, speed, and durability. I think it is a useful strategy for high-performance ionic actuator areas by using a new structured polyelectrolyte.*

Response: We deeply appreciate for reviewing our work with very positive opinions and valuable comments, which, we believe, can highly enhance the quality of this paper. Accordingly, we have revised the manuscript carefully regarding to your comments as follows:

Comment (1): *Before publication, I would like to suggest the authors add more evidence, discussion, and comparison about important details of the new membrane such as the pore size and size distribution, is there any big change compared with traditional nafion? The pore structure would be acting as effective ion channels for ions transportation and deformation caused by ions immigration.*

Response: We thank you for this comment. Accordingly, we would communicate as follows. We agree with reviewer that it could be straightforward if pore size and size distribution of ionic nano-channels would have been provided. Actually, we wanted, but were not able to do so due to two following reasons. First, Nafion micelles in nanoscale were too weak to form strong and stable porous membranes like solid porous materials such as graphdiyne (Fig. 1g, *Nat. Commun.* **9**, 2018, 752), carbon nitride (Fig. 6, *Nat. Commun.* **6**, 2015, 7258), and covalent organic framework (Fig. 3d, *Nat. Commun.* **13**, 2022, 390). This is the reason for using cryo-TEM to check the morphology of Nafion micelles in water (**Fig. 2a-e**). Second, when Nafion micelles were mixed with ionic liquid and cast membrane. Ionic nano-channels were not empty but filled with ionic liquid. Therefore, neither pure Nafion micelles nor H-membranes integrated with ionic liquid was able to be checked pore size by gas adsorption-desorption isotherm.

However, the configuration of Nafion molecules in the micelles was preserved because the membrane was cast at 90 °C, which was much smaller than the glass transition temperature of Nafion. Therefore, although we cannot check the pore size of ionic nano-channels, it could be sure that continuous conducting phase was formed. Accordingly, we would add the following sentence to the updated manuscript:

“During casting, the configuration of Nafion molecules in the micelles was preserved because the casting temperature (90 °C) was much smaller than the glass transition temperature of Nafion.^{56,57} Therefore, casting these as-made networked micelles with ionic hydrophilic surfaces covering hydrophobic cores can form the desired polyelectrolyte membranes and continuous conducting matrix of the hydrophilic side chains plasticized by EMImBF₄ greatly improves ionic conductivity.”

Comment (2): *In addition, I would like to suggest the authors redesign and replot their bionic synthetic idea about the hemostasis-inspired membrane in Fig1 (a-d), in order to let readers catch the idea quickly and easily.*

Response: We thank you for this comment. Accordingly, we have edited it as follow:

Fig. 1. Hemostasis-inspired electrolyte membrane for ionic artificial muscle. Simple illustrations for the structure of platelet and the hemostasis process: a. Blood vessel, b. Primary hemostasis, and c. Secondary hemostasis. Hemostasis-inspired procedure for preparing electrolyte membrane (H-membrane): d. Nafion micelle in water, e. Micelle aggregation, and f. Homostasis-inspired membrane. g. Application of H-membrane to ionic actuators for use in an inchworm-mimetic soft robot and a dynamic tensegrity system.

Authors' Responses to Reviewer #2

Reviewer's overall statement: *In this manuscript, the authors reported a hemostasis-inspired strategy to synthesize efficient polyelectrolyte membranes that have continuous conducting network suitable for electro-ionic actuator. And the developed artificial muscle shows great durability over 40 days under continuous actuation and much broader bandwidth below 10 Hz, which was applied to demonstrate an inchworm-mimetic soft robot and a kinetic tensegrity system. Overall, this manuscript was suitable to be accepted in Nature Communications after consideration of the following points:*

Response: We deeply appreciate for reviewing our work with very positive opinions and valuable comments, which, we believe, can highly enhance the quality of this paper. Accordingly, we have revised the manuscript carefully regarding to your comments as follows:

Comment (1): *The proportion of the dispersed system was determined by the calculation of Ra , but the experiments are lacking to verify whether the continuous conduction networks can also be formed at other proportions without sacrificing the mechanical strength.*

Response: We thank you for the comment and would communicate as follows. Nafion membranes have two separate components for ionic conductivity and mechanical strength. Ionically conductive domain is related to hydrophilic side chains integrated with ionic liquid and Mechanically elastic domain is related to hydrophobic tetrafluoroethylene backbones. Therefore, these two different domains depend on the portions and phase distributions of these two molecular components. However, there is no effective technique for controlling phase distributions of Nafion.

We anticipated that we could obtain continuous conducting phase *via* Nafion micelles with the surfaces of hydrophilic side chains. Normally, amphiphilic (containing both hydrophobic and hydrophilic components) molecules like Nafion can assemble into micelles if their distance parameter, Ra , is smaller than that of solvent.

The Ra value of Nafion is not available. However, according to the literature, this value should be between 2.69 – 20.09 MPa^{1/2}. We calculated solubility parameters of solvent mixtures of water and DMAc and did experiments with those with $Ra > 20.09$ MPa^{1/2}.

Proper Ra value of solvent is just a thermodynamic condition. Forming micelles is a kinetic process. Nafion molecules should disperse well at low concentrations before forming micelles. Therefore, we first diluted Nafion solution by DMAc, then gradually added water at high temperature. Finally, we let the mixture cool down slowly and kept it static for at least three days, providing proper conditions for micellar formation.

Here, we reported an optimized condition for making membrane with high ionic conductivity and acceptable mechanical strength (**Supplementary Table 1**). Slight change of water content around this condition could result in membranes with similar properties. However, our observation suggested that too small water leads to low ionic conductivity but strong membrane and too much water makes high ionic conductivity but too weak membrane.

Comment (2): *The effects of Nafion colloidal concentration on the mechanical strength and the conductivity of the as-prepared film are not supported by the existing experimental data.*

Response: We thank you for the comment and would communicate as follows. Mechanical strength and ionic conductivity of a membrane are directly affected by micellar rheology, which is governed by Nafion concentration and water portion in the solvent mixture. The effect of Nafion concentration can be seen clearly in **Fig. 2b-c** and **Supplementary Fig. 2**. Accordingly, lowering Nafion concentration resulted in fewer micellar networks, which led to membrane with less connection of non-conducting domain. As a result, membrane was weaker but had higher ionic conductivity. When we reduced Nafion concentration to 0.45%, the membrane was too weak that easily broke during actuator fabrication. We found that 0.7% Nafion generated good membrane compromising between mechanical strength and ionic conductivity. Therefore, we fixed this concentration and investigated the effect of water portion in the solvent mixture. After fixing the water content as reported in **Supplementary Table 1**, we characterized membrane properties in detail and showed in the manuscript. Thus, although we reported mechanical strength and ionic conductivity of membrane from one Nafion concentration, we identified a clear trend that lower Nafion concentrations made membranes weaker but higher conductivity and vice versa.

Comment (3): *The controllability of the as-prepared films must be strengthened by more experimental results.*

Response: We thank you for the comment and would communicate as follows. As mentioned in the response of previous comment, Nafion concentration and water portion in the solvent mixture directly affect micellar rheology, which influenced final membranes. The effect of Nafion concentration was discussed in the previous comment. Here, we would explain water portion in the solvent mixture.

Although all water-DMAc mixtures with Ra higher than $20.09 \text{ MPa}^{1/2}$ generated micelles, water content significantly changed micellar rheology. With increase in water content, the state of Nafion micelles evolved from strong gel (**Supplementary Fig. 5a**), weak gel, to precipitated dispersion like following (**Supplementary Fig. 5b-c**). This phenomenon can be explained by the fact that more water with higher Ra values resulted in fewer micellar networks as supported by **Fig. 2b-c** and **Supplementary Fig. 2**.

In addition, when ionic liquid was added, the viscosity of these dispersions increased rapidly due to the increase of particle sized as can be seen in **Fig. 2j** and **Supplementary Fig. 4**. Therefore, micellar concentration should also be considered in term of practical experiment that facilitates two factors, which are the homogenous dispersion of ionic liquid in the whole mixture and the ease casting of good membranes.

Finally, the ratio between water and DMAc not only affected micelle formation but also membrane generation related to solvent evaporation as discussed in the manuscript and can be seen in **Figs. 3a** and **3b**.

According to your comment, we would like to add the following figure to the **Supplementary** and more explanation in the main text as follows:

“The presence of water had an impact on both micellar rheology (as shown in **Supplementary Fig. 5**) and the ultimate membrane structure. The recipe provided in **Supplementary Table 1** represents an optimized condition that takes into account numerous parameters throughout the entire process, from preparing the Nafion solution to casting the membrane. Notably, during the

casting step, the arrangement of Nafion molecules within the micelles was maintained due to the casting temperature being significantly lower (90°C) than the glass transition temperature of Nafion.^{56,57}”

Supplementary Fig. 5. Effect of water content on micellar rheology. a. strong Nafion gel in low water content. **b.** precipitated Nafion micelles in high water content. **c.** Laser scattering of precipitated Nafion micelles in high water content. **d.** Laser propagation at high position.

Comment (4): *The mechanical properties of various films was displayed in Figure 3h-i, however, the yield strength is more important for the actuator than the tensile strength during the driving process, the data analysis method is suggested to adjust.*

Response: We thank you for the comment. Accordingly, we replaced tensile strength by yield strength. We revised the Figure and the updated manuscript as follows:

Fig. 3. Properties of H-membrane. **a.** 0 % and **b.** 8.5 % of DMAc in water affecting on the formation of membrane. **c.** Measurement setup for electrochemical impedance spectroscopy (EIS) and cyclic voltammetry (CV). **d.** Typical EIS spectra. **e.** Ionic conductivity. **f.** Typical CV curves. **g.** Specific capacitance. **h.** Typical tensile curves. **i.** Mechanical strength.

“Tensile curves are shown in **Fig. 3h** and **Supplementary Fig. 9b** were utilized to compute mechanical stiffness and strength as disclosed in **Fig. 3i**. Thereupon, H-membrane had lower Young modulus and yield strength than C-membrane.”

Comment (5): In addition to computational analysis, other experimental characterizations like FTIR and Raman can be also demonstrated to further prove the intermolecular effects of water and DMAc on the dispersion results.

Response: We thank you for the comment. Accordingly, we checked FTIR and Raman spectroscopy of Nafion in water and in DMAc. The data does not show much difference between two samples. Maybe, these two characterizations are not suitable in this case. However, the difference between two samples and the formation of micelles in water were clearly confirmed by Cryo-TEM, laser scattering, DLS, and zeta potential.

Comment (6): *The analysis about Figure 4f was mistakenly titled as Figure 4g in the manuscript.*

Response: We thank you for the comment. Accordingly, we have corrected it in the updated manuscript as follows:

“This advancement in bending deflection continued in the broad excitation frequency range from 0.01 to 20.0 Hz in **Fig. 4f** and in the applied voltage from 0.01 to 1.00 V in **Fig. 4g**. Most interestingly, **Fig. 4f** shows that the H-membrane actuator maintained tip displacement from 0.01 Hz to 1 Hz...”

Authors' Responses to Reviewer #3

Reviewer's overall statement: *In this study, the authors present the fabrication and characterization of a hemostasis-inspired IMPC actuator. The concept of a bio-inspired chemistry approach is fairly novel, and the IMPC does show a significantly faster response speed. However, the mechanical properties of the actuator do not seem to be better compared to existing literature, and the reviewer would like to request the authors to provide an elaborate discussion on this particular subject matter. The following comments should also be addressed:*

Response: We deeply appreciate for reviewing our work with very positive opinions and valuable comments. We believe that all comments were so insightful and helpful in enhancing the quality of the paper. For your comment about the mechanical properties, we would discuss as follows.

We mentioned about the relationship between mechanical strength and ionic conductivity of polymer electrolytes. In general, a polymer like Nafion has two components. One is non-conducting component like the backbone of Nafion; the other is conducting component like the side chains with sulfonate groups of Nafion (Fig. 2, *Chem. Rev. 2017, 117, 987–1104*). Due to their immiscibility, those two components form distinct phases. While non-conducting component forms rigid phase with some crystalline regions and is responsible for mechanical strength, conducting component form flexible phase and is responsible for conducting ions (Fig. 17, *Chem. Rev. 2017, 117, 987–1104*). Because each of these phases generates a separate property, the phase fraction and the phase distribution have direct and inverse effects on both mechanical strength and ionic conductivity. Although phase fraction and phase distribution directly correlate, for simply, we would discuss them separately.

First, we would discuss the effect of the phase fraction with two following examples. First example is Nafion. Water selectively mixes with conducting phase of Nafion. Higher water percentage expands conducting domain, which leads to increases in ionic conductivity (Fig. 34a, 34c, 35c, *Chem. Rev. 2017, 117, 987–1104*) and decreases in Young modulus (Fig. 41, 42a, *Chem. Rev. 2017, 117, 987–1104*). Second example is a polymer named poly(styrenesulphonate-b-methylbutylene). Its ionic conductivity improved according to the expansion of conducting phases due to increases in sulfonate groups (Fig. 2b *Nat. Commun. 4:2208, 2013*) and/or increases in ionic liquid (Fig. 2 *Nat. Commun. 1:88, 2013*). However, the modulus decreased with the increases in sulfonate groups and ionic liquid (Fig. 6c *Macromolecules 47, 4357–4368, 2014*).

Second, we would discuss the effect of the phase distribution here with two following examples. First example is Nafion. Water selectively mixes with conducting phase of Nafion. Higher water percentage expands conducting domain and makes this phase more connected, which leads to more separated non-conducting phase (Fig. 17, *Chem. Rev.* 2017, 117, 987–1104). This decreases in Young modulus (Fig. 41, 42a, *Chem. Rev.* 2017, 117, 987–1104). Second example is a triblock terpolymer/ionic liquid electrolyte membrane. Higher ionic liquid content caused the morphology to transit to less connected non-conducting phase, which led to the decrease of modulus (*Macromolecules* 2014, 47, 1090–1098).

These correlations lead to the inverse relationship between those two properties. This inverse relationship remains a dilemma (*J. Chem. Soc. Faraday Trans.*, 89(17), 3187-3203, 1993; *Nat. Mater.* 12, 452–457, 2013) and researchers have tried to find ways to compromise these two properties (*Annu. Rev. Mater. Res.* 43:503–25, 2013).

In every case, any increase in ionic conductivity causes mechanical strength to decrease. **In this work, we tried to maximize ionic conductivity without significantly sacrificing the mechanical strength.** For that, we first increased the fraction of conducting phase by increasing ionic liquid concentration in comparison with Nafion from 60% in conventional membrane to about 160% in our membrane (calculating from **Supplementary Table 1**). Second, we generated continuous conducting phase and isolated non-conducting phase *via* micelles. As a result, ionic conductivity increased by three times. Therefore, we believe that the reduction of mechanical strength in the H-membrane could be understandable and acceptable for largely bendable soft actuator at low input voltages.

For more clarification, we would like to add the following sentences to the revised manuscript:

“However, polyelectrolytes have received much less attention than their electrode counterparts, possibly due to a challenge of generating the continuous conducting network without significantly sacrificing the mechanical strength.^{24,35} **This challenge arises from the inherent distinction between two properties stemming from separate Nafion chains: ionic conductivity is derived from hydrophilic side chains, while mechanical strength mainly originates from hydrophobic non-conducting domains. Because these components are not compatible, they give rise to distinct phases. Consequently, the distribution and proportion of these phases exert both positive and negative influences on both mechanical strength and ionic conductivity.³⁹⁻⁴¹** Although several block-ionomers were reported with an expectation that their self-assembly could be a good solution,^{24,35} weak self-assembly force of block copolymers could not be suitable for governing their morphology of hundred-micrometer thick membranes needed in these practical applications.^{39”}

Comment (1): Line 43: It is suggested to use consistent (10-times) terminology.

Response: We thank you for the comment. Accordingly, we have edited in the updated manuscript as follows:

“..., which boosts the performance of electro-ionic soft actuators by **10-time** faster response and **36-time** higher tip-to-tip bending displacement.”

Comment (2): Lines 70-77: The authors argue that the mechanical properties of the conductive

network may decrease with conductive network improvement. It is suggested that the authors provide specific values for the "good" or "bad" mechanical stiffness. The reviewer suggests summarizing the literature results into a table.

Response: We thank you for the comment. Accordingly, we would respond as follow:

For your first suggestion, we agree with reviewer that it could be straightforward if we can “provide specific values for the "good" or "bad" mechanical stiffness”. Proper mechanical stiffness of electrolytes depends on many elements such as requirements of target applications (stiffer materials could be better for applications requiring strong forces; however, softer materials could be more appropriate for uses needing large deformations), stiffness of selected electrode materials (similar stiffness between these two materials could reduce internal stress and delamination), and the input voltage amplitude (insufficient stiffness of electrolytes could not resist the stress between two electrically connected electrodes, *Ionics* 22, 1259–1279, 2016). And, setting a standard mechanical stiffness of electrolytes should consider many other parameters such as ionic conductivity, electrode properties, actuator dimensions, actuation testing conditions, etc. Therefore, providing a standard for mechanical stiffness of electrolytes could be beyond the scope of a research paper like this manuscript. However, we agree that a standard set for electro-ionic actuators is crucial and need to be established. Maybe, some leading scholars in this field need to discuss and write a review paper about this issue.

According to your second suggestion, we have surveyed the literature. However, there have been few papers about polymer electrolytes in the field of ionic actuators. Because of the lack of standard criteria as mentioned previously, most paper reported data according to their specific focus and circumstance rather than general data on mechanical strength. And justifying proper mechanical strength should be considered together with electrode properties, actuator fabrication, and actuator applications as discussed in the previous paragraph.

Comment (3): *Figure 1 is very busy, most likely due to the embedded words, and it is suggested that the figure be further refined.*

Response: We thank you for the comment. Accordingly, we have edited it as in the updated manuscript.

Fig. 1. Hemostasis-inspired electrolyte membrane for ionic artificial muscle. Simple illustrations for the structure of platelet and the hemostasis process: a. Blood vessel, b. Primary hemostasis, and c. Secondary hemostasis. Hemostasis-inspired procedure for preparing electrolyte membrane (H-membrane): d. Nafion micelle in water, e. Micelle aggregation, and f. Homostasis-inspired membrane. g. Application of H-membrane to ionic actuators for use in an inchworm-mimetic soft robot and a dynamic tensegrity system.

Comment (4): Figure 2f-i caption: It is suggested that the authors elaborate on this particular caption with a discussion on the scattering light. The current caption does not explain the significance of the images.

Response: We thank you for the comment. Accordingly, we have edited in the updated manuscript as follows:

“f-i. **Laser scattering of Nafion dispersions in water and DMAc.**”

Comment (5): Figure 3c: It is suggested that the proper labels are included, similar to the following figures.

Response: We thank you for the comment. Accordingly, we have edited in the updated manuscript as follows:

“c. Measurement setup for electrochemical impedance spectroscopy (EIS) of electrolyte membranes and cyclic voltammetry (CV) of ionic actuators.”

Comment (6): Please explain why the conventional (C-membrane) shows a higher temperature dependence, as shown in Figure 3d.

Response: We thank you for the question. Accordingly, we would explain as follows. EIS data presented in Fig. 3d reflected and were used to calculate ionic conductivity. Roughly, the Z' value of the lowest point of EIS spectrum is linear to resistance, reciprocal of ionic conductivity. Ionic conductivity is not an absolute constant. It depends on the correlation between the receiving and dissipating energy of ions in membranes. Dissipating energy of ions should pay to overcome friction of the environment. Here, there are two types of receiving energy, which are from heat and electrical field of applied potential. For this test, electrical field is a constant (fixed applied potential during measurement).

For C-membrane, because the dissipating energy is very high, the temperature increase provided much thermal energy for ions to overcome high friction, which caused fast ion movement and sharp decrease in Z' value (increase in ionic conductivity). Despite the large gaps (big absolute values) among different temperatures, this dependence actually follows to Arrhenius law as can be seen in Fig. 3e.

Comment (7): Lines 242-261: The authors mentioned that PEDOT:PSS was coated onto the membrane as an electrode. Please comment on the adhesion of nafion/PEDOT:PSS between the C- and H-membrane.

Response: We thank you for the comment. Accordingly, we have edited the updated manuscript as follows:

“These values increased from 1.7 to 4.5 times when excitation frequencies ranged from 0.1 to 5.0 Hz. These enhancements in specific capacitances mean that, with the same electrodes and testing conditions, H-membrane had more ions going to electrodes than C-membrane, especially at high frequencies. This could be due to two following reasons. First, H-membrane with higher ionic conductivity should have supplied more ions to neutralize more charges in electrodes than C-membrane. Therefore, the improvement of specific capacitance and ionic conductivity are inter-supported and originated from more efficient continuous conducting phase in the H-membrane. Second, H-membrane with stronger bonding to PEDOT:PSS electrodes should have facilitated more ion movement between electrolyte and electrodes than C-membrane because the surfaces of the former membrane made of Nafion micelles in water had much more sulfonate groups than those of the latter membrane made of Nafion dispersion in DMAc. The detailed geometry of the polyelectrolyte specimen for the tensile test are shown in Supplementary Fig. 8a. Tensile curves are shown in Fig. 3h and Supplementary Fig. 8b were utilized to compute mechanical stiffness

and strength as disclosed in **Fig. 3i.**”

Comment (8): Line 256: "H-membrane had a lower Young's modulus and tensile strength than C-membrane. This suggests that the H-membrane had fewer connected non-conducting domains." The reviewer has a minor concern about the above statement; the lower mechanical properties should not be directly correlated to a better connected network.

Response: We thank you for the comment. By following your comment, we removed the part.

Comment (9): Line 260: "In other words, normally PEDOT:PSS electrodes are much stiffer than the polyelectrolyte core layer." Please provide proper references and specific values.

Response: We thank you for the comment. Accordingly, we have added a reference and specific values as in the updated manuscript:

"In other words, normally PEDOT:PSS electrodes are much stiffer than the polyelectrolyte core layer. It was reported that cast PEDOT:PSS film with 30-micrometer thickness had Young modulus of about 2 GPa and tensile strength of about 43 MPa.⁵⁸

Comment (10): Line 263: "well-connected non-conducting domains for ideal mechanical stiffness." Following the previous comment, the reviewer would like to raise the question of the authors justifying the "ideal mechanical stiffness" as the H-membrane has much worse mechanical properties, both in strength and strain. Also, it is suggested that the authors provide mechanical testing results on both PEDOT:PSS coated and non-coated samples for comparison.

Response: We thank you for the comment. Accordingly, we would respond as follow:

For your comment, a certain range of mechanical stiffness of electrolytes is suitable for ionic actuators. Too rigid electrolytes generate very small deformation; however, too soft electrolytes lead to very small blocking forces. Therefore, by "ideal mechanical stiffness", we meant that a *compromised* stiffness. However, the word "ideal" could cause misunderstanding. Hence, we have edited this sentence as in the updated manuscript:

“All data of these three characterizations indicate that H-membrane had continuous conducting phases for high ionic conductivity and well-connected non-conducting domains for **compromised** mechanical stiffness, which are basically required for high-performance ionic soft actuators.”

According to your suggestion on mechanical test, we have conducted more experiment and provided more data of tensile tests on PEDOT:PSS coated samples in the updated manuscript and the updated Supplementary Information as follows:

Supplementary Fig. 10. Mechanical properties of actuators. a. Stress-strain curves of H-actuator. **b.** Stress-strain curves of C-actuator. **c.** Mechanical properties.

We would add one sentence for this figure in the main text:

“Therefore, actuators derived from these two membranes had similar mechanical strength as showed in **Supplementary Fig. 10**. All data of these three characterizations indicate that H-membrane had continuous conducting phases for high ionic conductivity and well-connected non-conducting domains for **compromised** mechanical stiffness, which are basically required for high-performance ionic soft actuators.”

Comment (11): Line 265: “The mechanical stiffness of the polyelectrolyte can be tailored by changing the ratio between water and DMAc, based on the functionally antagonistic solvent procedure.” This statement should be removed unless the authors provide experimental results.

Response: We thank you for the comment. Accordingly, we have removed this sentence in the updated manuscript as follows:

Comment (12): Figures 4h & i: Is it possible to also add the long-term performance of the C-membrane to these graphs for comparison?

Response: We thank you for the comment. Accordingly, we have checked and added the long-term durability data of C-membrane actuator. For proper comparison, we measured bending displacement at 3 Hz. However, at 0.3 V, C-membrane had very small displacement as can be seen in the inset of **Fig. 4g**, which made difficulty in monitoring the sample. Therefore, we increased the applied voltage to 1.5 V for clear visible displacement. This actuator showed poor durability less than one day. One reason could be due to fast bending at high frequency of 3 Hz and poor bonding between electrodes and electrolyte, which lead to delaminating of these two components and rapid decrease in displacement. Because of its short time and different applied voltage, we would like to add this figure in the Supplementary as follows.

Supplementary Fig. 17. Durability of C-membrane actuator at 1.5 V and 3.0 Hz, normalized to the initial bending displacement.

We would add one sentence for this figure in the main text:

“The displacement values were normalized with the averaged displacement over 40 days. As presented in **Fig. 4h** and **Supplementary Fig. 16**, the H-membrane actuator exhibited excellent cyclic stability over 42 days, indicating about 11 million cycles in the continuous harmonic excitation. **The durability of the C-membrane actuator under continuous sinusoidal excitation with peak voltage of 1.5 V and the frequency of 3.0 Hz is relatively poor as shown in Supplementary Fig. 17.**”

At this stage, we would like to mention that the reviewers’ comments were so highly pertinent and helpful that we can enhance the quality of the article and deepen the overall level of our scientific understanding. We enormously appreciate all reviewers for the careful review, and we hope that the paper with such significant improvement is now acceptable for publication. All of the changes in the manuscript are marked in red.

REVIEWER COMMENTS

Reviewer #1 (Remarks to the Author):

The paper is well revised and suggested to be accepted.

Reviewer #2 (Remarks to the Author):

The manuscript can be accepted under this modification version.

Reviewer #3 (Remarks to the Author):

In this study, the authors present the novel fabrication and thorough experimental characterization of a hemostasis-inspired electro-ionic soft actuator. The membrane and combined actuator are comprehensively characterized in a series of mechanical, microstructure, and performance tests. The developed actuator demonstrates greatly improved actuation speeds, maximum displacement and repeatability compared to their baseline, a conventionally fabricated ionic-polymer actuator. The reviewer acknowledges the revisions made to address the previous comments and would recommend this study to be published in Nature Communications. The following are some minor comments on the manuscript.

1. Results and Discussion, Micelle properties. In the paragraph encompassing lines 204 to 219, the authors refer to and discuss a graph labeled Fig. 2k. The labeling of images in Figure 2. may have changed in the revisions and discussed figure does not exist. It is likely referring to Fig. 2j.
2. Results and Discussion, Membrane properties. The authors state that “actuator sandwiched with PEDOT:PSS electrodes usually has minor contribution to the total mechanical stiffness in the actuator level...Therefore, actuators derived from these two membranes had similar mechanical strength...” However, in Supplementary Fig. 10, there is a notable reduction in the mechanical stiffness and modulus when comparing the H-actuator to the C. The reviewer recommends this section to be changed to have consistent phrasing when analyzing the mechanical properties.
3. Results and Discussion, Actuation Performance. The electro-ionic actuator used for this study can be highly influenced by the temperature and humidity conditions. The reviewer suggests briefly mentioning the average temperature and relative humidity levels, and any steps taken to prevent fluctuations in the testing conditions.

Authors' Responses to Reviewers

Authors' Responses to Reviewer #1

The paper is well revised and suggested to be accepted.

Response: We deeply appreciate your review of our work and suggestion of acceptance.

Authors' Responses to Reviewer #2

The manuscript can be accepted under this modification version.

Response: We deeply appreciate your review of our work and suggestion of acceptance.

Authors' Responses to Reviewer #3

Reviewer's overall statement: In this study, the authors present the novel fabrication and thorough experimental characterization of a hemostasis-inspired electro-ionic soft actuator. The membrane and combined actuator are comprehensively characterized in a series of mechanical, microstructure, and performance tests. The developed actuator demonstrates greatly improved actuation speeds, maximum displacement and repeatability compared to their baseline, a conventionally fabricated ionic-polymer actuator. The reviewer acknowledges the revisions made to address the previous comments and would recommend this study to be published in Nature Communications. The following are some minor comments on the manuscript.

Response: We deeply appreciate your review of our work and suggestion of acceptance.

Comment (1): Results and Discussion, Micelle properties. In the paragraph encompassing lines 204 to 219, the authors refer to and discuss a graph labeled Fig. 2k. The labeling of images in Figure 2. may have changed in the revisions and discussed figure does not exist. It is likely referring to Fig. 2j.

Response: We thank you for the comment. Accordingly, we have edited in the updated manuscript as follows:

“As expected, adding EMImBF₄ to the as-prepared micellar dispersion caused aggregations as showed in Fig. 2f-j and Supplementary Fig. 4.”

Comment (2): Results and Discussion, Membrane properties. The authors state that “actuator sandwiched with PEDOT:PSS electrodes usually has minor contribution to the total mechanical stiffness in the actuator level...Therefore, actuators derived from these two membranes had similar mechanical strength...” However, in Supplementary Fig. 10, there is a notable reduction in the

mechanical stiffness and modulus when comparing the H-actuator to the C. The reviewer recommends this section to be changed to have consistent phrasing when analyzing the mechanical properties.

Response: We thank you for the comment. Accordingly, we would remove this statement from the updated manuscript.

Comment (3): Results and Discussion, Actuation Performance. The electro-ionic actuator used for this study can be highly influenced by the temperature and humidity conditions. **The reviewer suggests briefly mentioning the average temperature and relative humidity levels**, and any steps taken to prevent fluctuations in the testing

conditions.

Response: We thank you for and totally agree with the comment. Actually, during the experiment, we recorded the humidity and outdoor temperature.

We would add this information in the updated manuscript as follow:

“Bending displacement and blocking force were measured by a laser sensor (LK-H080, KEYENCE) and a load cell (LVS-5GA, KYOWA), respectively (**Supplementary Fig. 12**). During checking actuators, the temperature in the testing room was maintained between 18 and 20 °C and the humidity was between 60 and 90 %.”

In addition, we would like to add the following supplementary figure in the updated manuscript:

Supplementary Fig. 17. Durability of A-membrane actuator, humidity, and outdoor temperature. The temperature in the testing room was maintained between 18 and 20 °C. The humidity was between 60 and 90 %.

REVIEWERS' COMMENTS

Reviewer #3 (Remarks to the Author):

All comments have been addressed and the manuscript can now be accepted.